# Host Range Evolution of Potyviruses: A Global Phylogenetic Analysis

**DOI:** 10.3390/v12010111

**Published:** 2020-01-16

**Authors:** Benoît Moury, Cécile Desbiez

**Affiliations:** INRAE, Pathologie Végétale, F-84140 Montfavet, France; cecile.desbiez@inra.fr

**Keywords:** potyvirus, potyviridae, ancestral reconstruction, discrete character, host jump, host shift, host range expansion

## Abstract

Virus host range, i.e., the number and diversity of host species of viruses, is an important determinant of disease emergence and of the efficiency of disease control strategies. However, for plant viruses, little is known about the genetic or ecological factors involved in the evolution of host range. Using available genome sequences and host range data, we performed a phylogenetic analysis of host range evolution in the genus *Potyvirus*, a large group of plant RNA viruses that has undergone a radiative evolution circa 7000 years ago, contemporaneously with agriculture intensification in mid Holocene. Maximum likelihood inference based on a set of 59 potyviruses and 38 plant species showed frequent host range changes during potyvirus evolution, with 4.6 changes per plant species on average, including 3.1 host gains and 1.5 host loss. These changes were quite recent, 74% of them being inferred on the terminal branches of the potyvirus tree. The most striking result was the high frequency of correlated host gains occurring repeatedly in different branches of the potyvirus tree, which raises the question of the dependence of the molecular and/or ecological mechanisms involved in adaptation to different plant species.

## 1. Introduction

Host range of a parasite is defined as the set of organisms where the parasite can perform, at least partly, its infection cycle. The number and diversity of host species are frequently used to measure a parasites’ host range. Host range is an important notion in parasitology, since it is linked to risks of disease emergence [1] and has important consequences on the implementation and efficiency of disease control strategies including epidemiological surveillance, quarantine measures, vaccination, and deployment of parasite-resistant crop varieties in agricultural landscapes [2,3,4].

In phytopathology, parasites’ host range has been used for decades for taxonomic purposes, both at parasite inter- and intraspecies levels. Host range is determined both by factors intrinsic to the parasite that determine its fitness in different hosts and by extrinsic factors related to parasite ecology and epidemiology [5]. Overall, there is little knowledge on determinants of the host species range of plant parasites by contrast with determinants of infection at the plant intraspecific level that have been the subject of numerous studies, particularly about the ability of a parasite to infect plant genotypes with specific resistance genes [5]. For plant viruses, relatively few studies are available on host range genetic determinants or on host range changes that have occurred during virus evolutionary history [6,7,8,9,10,11,12,13,14,15,16].

We used a phylogenetic approach to unravel the evolution of the host species range in the genus *Potyvirus*. Given its large number of species (175 member species and 31 tentative species), this genus is favorable for this kind of approach [17]. Collectively, its members have a broad host range that encompass 57 plant families, mostly in the division Magnoliophyta (i.e., angiosperms or flower plants) [18]. However, each potyvirus species usually has a narrow host range, with a few exceptions like *Bean yellow mosaic virus*, *Turnip mosaic virus,* and *Watermelon mosaic virus* (WMV) which infect hosts in at least 12 plant families each [18]. Phylogenetic analyses showed that the diversification of the genus *Potyvirus* is quite recent, since it began circa 7000 years ago, probably in the greater region around the Fertile Crescent of the Middle East, where agriculture emerged about 9500 years ago [19]. This suggests that potyviruses have undergone numerous host range changes in a relatively short amount of time and that crop plants have contributed substantially to the diversification of potyviruses.

Our objectives were to unravel the nature and frequency of host range changes that occurred during the evolution of potyviruses and the distribution of these changes among the potyvirus and plant diversity.

## 2. Materials and Methods

### 2.1. Collating Host Range Data

Host range data of potyvirus species were obtained from two bibliographic sources, which have established quite exhaustive lists of host and non-host plant species for each potyvirus species [18,20]. In these sources, non-hosts had been determined by laboratory inoculation experiments, whereas hosts corresponded either to naturally or experimentally infected species. Local-lesion hosts had been considered as hosts. In case of disagreement between the sources, i.e., when a given plant species was considered to be either host or non-host of a given potyvirus species, we considered it to be a host, since this discrepancy is likely due to the use of different plant genotypes and/or virus isolates in different studies.

### 2.2. Phylogenetic Analyses

For each potyvirus species for which we could obtain host range data, we checked whether full genome or near full genome sequences were available in databanks. Genome data were obtained for 57 potyvirus species, to which we added a partial genome of *Iris mild mosaic virus* (IMMV; accession JF320812; excluding most of the P1 cistron) and a complete genome of *Amaranthus leaf mottle virus* (AmLMV) obtained in our laboratory (accession number MN709786). Sequences of three members of the genus *Rymovirus*, the sister group of the genus *Potyvirus* in the family Potyviridae, were added as outgroups to root the phylogenetic tree and to help reconstruct host range changes. These were *Hordeum mosaic virus* (HoMV), *Agropyron mosaic virus* (AgMV), and *Ryegrass mosaic virus* (RgMV).

Nucleotide or amino acid sequences of potyviruses and rymoviruses were aligned using MAFFT (Multiple Alignment using Fast Fourier Transform) version 7 [21]. Then, the best substitution model was selected by maximum likelihood using the MEGA software version 7 [22]. Finally, phylogenetic analyses were conducted using the selected substitution model and the maximum likelihood algorithm implemented in MEGA version 7. One hundred bootstraps were performed to evaluate the reliability of the tree topology.

### 2.3. Determination of Ancestral Host/Non-Host States in Potyvirus Phylogenetic Tree

For each plant species, by combining the obtained virus phylogeny (topology and branch lengths) and the host and non-host data, we assigned to each internal branch *i* of the virus phylogenetic tree the probability *P_i_* that the given plant species was a host of the corresponding potyvirus ancestor. Hence, the probability that the given plant species was non-host of that potyvirus ancestor is (1 − *P_i_*). For this, we compared the likelihood of two models using the ‘ace’ function [23] in the ‘ape’ package [24] of the software R version 3.5.3 [25]. The equal-rates (ER) model considers that the evolution from host to non-host and from non-host to host are equally probable, whereas the all-rates-different (ARD) model considers two different evolutionary probabilities, one for the host-to-non-host transition and the other one for the non-host-to-host transition. Since there are only two possible modalities among our data (host and non-host), the ER and ARD models differ by one parameter only and their likelihoods can be compared using a Chi-squared test with one degree of freedom. Then, the obtained probabilities *P_i_* associated to the different branches of the phylogenetic tree allowed identifying the branches corresponding to host range changes. When changes corresponded to internal branches in a clade containing host/non-host data for a single virus species and no data was available for the other viruses in the same clade, the change was attributed to the terminal branch corresponding to that virus.

To evaluate the respective impacts of (i) the total numbers of viruses for which a given plant species is host or non-host and (ii) their distribution in the potyvirus phylogenetic tree on the host or non-host state of that plant species for the most recent common ancestor (MRCA) of potyviruses, we performed permutation tests. For these, host range data were randomly permuted among the virus species before re-inferring the probability that the plant species was a host of the potyvirus MRCA. Ten thousand permutations were performed to obtain probability values.

The Monte Carlo method was used to test if there was a significantly low or high frequency of host gains, host losses, or total host changes in the potyvirus phylogenetic tree depending on the plant family or the potyvirus clade. The Monte Carlo method was also used to test if, compared to distributions expected at random, there was significantly lower or higher frequencies of (i) host change co-occurrences (i.e., change of the host status of several plant species inferred on the same branch of the potyvirus phylogenetic tree) and (ii) repeated host change co-occurrences, i.e., the same host change co-occurrence found on several branches of the potyvirus phylogenetic tree. In all cases, the probability distribution used for simulations was proportional to the branch lengths or to the sum of branch lengths when groups of plant or virus species were analyzed. When sums of branch lengths were taken into account, we excluded the branches corresponding to viruses for which no host range data was available.

## 3. Results

### 3.1. Building a Potyvirus and Rymovirus Phylogenetic Tree

To identify host range changes during the evolution of potyviruses, we first established the phylogenetic relationships between 62 virus species for which genome and host range data were available: 59 *Potyvirus* species and three *Rymovirus* species used as outgroups. Reliable alignments of the polyprotein amino acid sequences were obtained, except for the P1 region. In contrast, alignments were uncertain for several genome regions using nucleotide sequences. The best-suited substitution model, i.e., the LG (Le-Gascuel) model [26] with gamma-distributed rates among sites, invariant sites, and including amino acid frequencies, was used to infer virus phylogeny with the amino acid alignments (Figure 1). In spite of the use of a different set of potyvirus species, the tree topology was consistent with that obtained by Gibbs and Ohshima [19] and the same potyvirus groups were monophyletic in both studies (Figure 1).

### 3.2. Frequent Host Range Changes during Potyvirus Evolution

For each plant species, we used the topology and branch lengths of the phylogenetic tree together with the host/non-host data to infer the ancestral states in the virus tree. We showed that less than 2% of random simulations showed a probability above 0.65 that a given plant species was a host or a non-host for the potyvirus MRCA corresponding to branch number 64 (Figure 1) (i.e., *P*_64_ > 0.65 or (1 − *P*_64_) > 0.65) (Appendix B). Hence, we considered that the host or non-host status of the potyvirus MRCA was reliable if *P*_64_ > 0.65 or (1 − *P*_64_) > 0.65. According to this criterion, no reliable inference could be obtained for plant species in which the host/non-host status is known for less than 12 of the 62 viruses (data not shown). Among the 66 remaining plant species (Appendix A), a reliable inference of the state of potyvirus MRCA was obtained for 39 plant species (Table 1).

There were large differences in the plant resistance spectrum and in the virus host range breadth in the remaining dataset of 39 plant species. The resistance spectrum varied from 8% to 95% of the potyviruses (mean of 62%) across plant species. Taking into account the potyviruses for which host/non-host data were available for at least 12 plant species, the host range breadth varied from 30% to 93% of the plant species (mean of 62%) across potyviruses. For most (35 of 39) plant species, the ER model, which assumes the same rate for host-to-non-host and non-host-to-host transitions, was preferred over the ARD model, which considers two different rates for these two kinds of transitions, according to likelihood ratio tests (LRTs), or was the only feasible model because the ARD model did not converge (Table 1). For the four remaining plant species, the ARD model fitted the data slightly better than the ER model according to LRTs, with probability values close to the 5% threshold (3.4% to 4.7%; Table 1). However, under the ARD model, the state probabilities *P_i_* of being host or non-host associated to many internal branches in the virus tree were close to 50%, which precluded reconstruction of host range changes. Consequently, since the LRTs were close to the 5% threshold, we also used the host/non-host state inferences obtained with the ER model for these four plant species.

Using the state probabilities associated to each branch in the tree, we could reliably infer the host range changes and identify the corresponding branches in the potyvirus phylogeny for 31 of the 39 plant species. The state of many internal branches in the potyvirus tree could not be inferred reliably for *Nicotiana tabacum*, with *P_i_* values close to 50%, probably because there were too many state changes (at least 15) for this plant species. Consequently, *N. tabacum* was discarded from following analyses. For seven other plant species (*Avena sativa*, *Capsicum annuum*, *Chenopodium amaranticolor*, *Nicotiana benthamiana*, *N. megalosiphon*, *Phaseolus vulgaris,* and *Vigna unguiculata*), there was an uncertainty for the number and sometimes the direction of host/non-host state transitions and for the identity of the corresponding branches in a particular part of the phylogenetic tree.

Host status changes that could not be attributed reliably to a precise branch of the potyvirus phylogenetic tree were excluded from further analyses, except in the comparison of the number of host status changes among plant families. In that latter case, the mean number of host status changes in the different plausible evolutionary scenarios was considered (Table 1).

Among the 38 plant species for which a reliable inference of host or non-host status of the potyvirus MRCA was obtained, most (34/38; 89%) belonged to only six botanical families: Amaranthaceae, Poaceae, Brassicaceae, Solanaceae, Cucurbitaceae, and Fabaceae (Table 2). Most plant species (30/38; 79%) were inferred to be non-hosts for the potyvirus MRCA. The eight other plant species, which are inferred to be hosts for the potyvirus MRCA, include wild plant species frequently used as laboratory plants (three *Nicotiana* spp. and two *Chenopodium* spp.) and three members of the Fabaceae (*Lathyrus odoratus*, *Lupinus albus* and *Trigonella foenum-graecum*). Permutation tests showed that the host/non-host state of the potyvirus MRCA was primarily influenced by the total number of viruses for which the plant species had the same state (Table 1; column 5 ‘Permutation test’). However, for ten plant species inferred to be non-host of the potyvirus MRCA, the total number of viruses for which the plant species is non-host was not sufficient to explain this inference and the distribution of viruses for which the plant species is host or non-host on the tree was also important.

On average, for each plant species, 4.61 host/non-host state changes were inferred with our set of potyviruses. Non-host-to-host transitions were about twice more frequent than the opposite transitions (3.12 versus 1.49 per plant species on average) (Table 1). For nine (or 10 depending on the host status reconstruction for *Avena sativa*) plant species, both kinds of transitions were inferred to have occurred during potyvirus evolution. However, for most plant species, a single type of transition was inferred. Consequently, for plant species inferred to be non-hosts of the potyvirus MRCA, means of 3.95 host gains and 0.48 host loss were predicted, whereas means of zero host gain and 5.25 host losses were predicted for plant species inferred to be hosts of the potyvirus MRCA.

### 3.3. Distribution of Host Range Changes across Plant Families and Potyvirus Clades

In spite of some differences in the numbers of inferred host gains, host losses, or total host changes across plant families or potyvirus clades (Appendix A), these differences were almost never significant statistically when considering change frequencies, i.e., numbers of changes per total branch length. The single exception was the significantly higher frequency of host changes in the Amaranthaceae than in the Poaceae (Kruskal–Wallis multiple test; *p*-value < 0.05) (Appendix A). Accordingly, the frequencies of host changes, host gains, or host losses inferred in the six main analyzed plant families (Table 2) were within the 95% confidence intervals (CIs) of random Monte Carlo simulations. Monte Carlo simulations showed also only two significant departures from random expectations for host gains, host losses, or total host changes inferred among the 12 main potyvirus clades (Figure 1). First, no host gain was inferred in clade 4 (including CTLV and KoMV) whereas from 1 to 9 (95% CI) gains would be expected according to Monte Carlo simulations. Second, 11 host gains were inferred in clade six (including *Turnip mosaic virus* (TuMV), *Narcissus yellow stripe virus* (NYSV), and *Narcissus late season yellows virus* (NLSYV)) whereas from one to nine (95% CI) gains would be expected according to Monte Carlo simulations. In both cases, the 99% CIs of Monte Carlo simulations included the inferred value.

According to their position on the virus phylogenetic tree, most host range changes are recent. Indeed, about 3/4 of the changes (115/156; 74%) were placed on terminal branches, whereas 26% were placed on internal branches (Appendix A). Similar imbalances were also observed for host gains (73 and 35 on terminal and internal branches, respectively) and host losses (42 and six on terminal and internal branches, respectively). Distributions of host changes, gains, or losses in terminal and internal branches were however not significantly different from what could be expected at random if they occurred proportionally to the branch lengths (Monte Carlo simulations; *p* > 0.05). Taking into account the host range changes placed on terminal branches only, 1.24 host gain and 0.71 host loss per potyvirus species were inferred on average. The majority of the 35 host gains inferred onto internal branches of the potyvirus phylogenetic tree (21/35; 60%) occurred on only four branches. These were branch numbers 82 (grouping *Bean yellow mosaic virus* (BYMV) and *Clover yellow vein virus* (ClYVV); 11 inferred host gains), 100 (clade seven or *Pepper veinal mottle virus* (PVMV); three host gains), 104 (clade eight or *Papaya ringspot virus* (PRSV); three host gains), and 105 (grouping clade nine or *Peanut mottle virus* (PeMoV) and clade 10 or *Bean common mosaic virus* (BCMV); four host gains) (Appendix A).

We also analyzed the co-occurrence of host changes, host gains, or host losses. Compared to the 95% CI of Monte Carlo simulations, there was a significant excess of co-occurrence of host gains (or of total host changes) for seven terminal branches and two internal branches of the potyvirus tree (Figure 1 and Appendix A).

Finally, we observed a large number of repeated (parallel) host gain co-occurrences (Table 3). Considering all possible pairs of plant species, a total of 119 host gain co-occurrences were observed on at least two different branches of the potyvirus phylogenetic tree. This is significantly higher than what would be expected by chance. Indeed, among 10,000 simulations where the host gains were randomly distributed among the tree branches proportionally to their length, a maximum of 23 (mean of eight) repeated co-occurrences of host gains were obtained. Among the 119 repeated co-occurrences of host gains, 43 (36%) corresponded to plant species of the same family and 55 (46%) corresponded to plant species sharing the same continent of origin, America, Africa, or Eurasia/Mediterranean Basin (Table 3). These frequencies are significantly higher than expected by chance considering the plant species distribution among families (*p*-value = 0.0047) or among continents of origin (*p*-value = 0.0476) by comparison with 10,000 Monte Carlo simulations. By contrast, repeated co-occurrence of host losses, or of one host gain and one host loss were much less frequent with nine and two cases, respectively (Appendix A).

## 4. Discussion

Given the paucity of data on host range evolution in plant viruses, we conducted a global analysis to unravel its main tendencies in potyviruses, a wide group of RNA viruses. Using host range compendia, we could accurately infer, in the potyvirus phylogenetic tree, the changes in the capacity to infect 38 plant species, i.e., the changes in potential host range. In nature, potyviruses are transmitted horizontally by aphids, i.e., insect vectors from superfamily Aphidoidea. As potyviruses are easily mechanically transmissible, their host ranges can be tested in laboratory conditions without biases related to host or virus specificity of the aphid vectors. However, virus host range evolution in natural conditions can be shaped by extrinsic factors not taken into account here [5].

This study revealed the ubiquity of changes in the potential host range during potyvirus evolution. Given the recent age of the potyvirus radiation, which started about 7000 years ago and is consequently largely posterior to the speciation of the studied plant species, these host changes correspond to host jumps rather than to host–virus co-divergences. This time frame of host changes is quite similar to other virus species, genera, or families. Indeed, the ages of family Luteoviridae, genus *Sobemovirus*, and species *Cucumber mosaic virus* (family Bromoviridae), *Citrus tristeza virus* (family Closteroviridae), *Tomato yellow leaf curl virus* (family Geminiviridae), and *Turnip yellow mosaic virus* (family Tymoviridae) were estimated to be <12,000 years (reviewed in [27]). The age of genera *Begomovirus*, *Mastrevirus,* and *Tobamovirus* was more ancient, with at least partial evidence of host–virus co-divergence for the latter [8].

Eighty percent (30/38) of analyzed plant species were inferred to be non-host of the potyvirus MRCA and for these plant species 3.95 host gains and 0.48 host loss were inferred on average, among a set of 59 potyviruses. This suggests that overall potyviruses have undergone host range expansions, i.e., an increase in the number of potential hosts, rather than host shifts, i.e., concurrent host gains and losses. Taking into account all 38 plant species, averages of 3.12 host gains and 1.49 host loss were estimated. These host change frequencies are underestimated for several reasons. First, some host/non-host data are missing in our dataset. Indeed, when a virus is first described on a specific plant, plants from the same genus or family are frequently over-represented in subsequent host range studies, and very distant species (e.g. monocots for a dicot-infecting virus) are more rarely tested. Second, some plant species may be wrongly considered as non-hosts because of the choice of particular plant genotypes and/or virus isolates to perform the tests. Third, the host change inference is largely based on parsimony principles which minimize the numbers of host changes per branch length (i.e., time unit), and fourth, some plant species may have been excluded from our analyses because too large numbers of host changes have precluded their reliable inference. The number of plant species corresponding to the latter case is difficult to estimate because the lack of reliable inference may either be due to an excessive number of host changes, an excessive number of host/non-host missing data, or both.

The inferred host gains, host losses, and total host changes were fairly distributed among the main potyvirus clades and among the main plant families examined. However, it should be emphasized that most plant species belonged to only six families corresponding to important groups of crop species, what may introduce a bias.

The majority (68%) of host gains were placed on the terminal branches of the potyvirus phylogenetic tree, meaning that they occurred after the potyvirus diversification into separate species. Among the 73 gains in potential host range that were inferred on terminal branches of the potyvirus tree, 27 cases (37%) correspond to natural host gains, i.e., the virus was found to infect the plant in field or natural conditions. This high relative frequency of host gains in terminal branches is compatible with a role of host jumps on the specialization and isolation of virus populations in new plant species environments and subsequently on potyvirus speciation [28]. Host gains inferred on internal branches of the potyvirus phylogenetic tree were scarcer and concentrated on a few branches. While the global radiation of potyviruses took place circa 7000 years ago, the diversification of some clades into species was estimated around 3600 years ago [28,29], and intraspecific diversification probably took place only a few centuries ago [30,31]. Indeed, many host–virus encounters that could lead to host jumps took place less than 500 years ago with the intensification of intercontinental exchanges of plants [30,32].

Given the small number of plant species analyzed in the present study and the bias in favor of crop species, it is unlikely to help identifying the plant species or families from which the genus *Potyvirus* originated. Accordingly, 80% (31/39) of the analyzed plant species were inferred to be non-hosts of the potyvirus MRCA (Table 1). Five of the remaining species are laboratory plants chosen in many studies because of their susceptibility to many viruses, inducing either local lesions or systemic infections (*Chenopodium amaranticolor*, *C. quinoa*, *Nicotiana benthamiana*, *N. clevelandii,* and *N. sylvestris*). The inference that they were hosts for the potyvirus MRCA may reflect the fact that they are hosts of most of the analyzed potyviruses (from 73% to 85%). The three other plant species (*Lathyrus odoratus*, *Lupinus albus,* and *Trigonella foenum-graecum*) belong to the same family Fabaceae and originate from the Near East or the Mediterranean Basin. However, the accuracy of their inference as hosts of the potyvirus MRCA should be taken with care because relatively few viruses of our dataset have been tested on these plants (13 or 14). Gibbs and Ohshima [18] assumed that the *Potyvirus* genus emerged from wild monocotyledonous plants around the Fertile Crescent of the Middle East. This assumption was based on the topology of the potyvirus phylogenetic tree. The sister group of potyviruses, genus *Rymovirus*, and the two “early-branching” clades of potyviruses (clades 11 or *Sugarcane mosaic virus* (SCMV) and 12 or *Onion yellow dwarf virus* (OYDV)) contain viruses that infect monocot plants (in families Poaceae or Liliaceae) that were domesticated in the broad geographical region spanning from Southwest Eurasia to North Africa. However, early-branching lineages do not necessarily signify ancestral traits as emphasized by Crisp and Cook [33] and intuitive assertion of ancestry from phylogenetic trees may be misleading. In our analyses, the five monocots (*Allium cepa* and four species of the Poaceae) were inferred to be non-hosts for the potyvirus MRCA but were only tested against a small subset of viruses (12 to 15, except *Zea mays*, a New World Poaceae, which was tested against 25 viruses). It remains also possible that, after jumping to crop plants, the ancestor of potyviruses had lost the capacity to infect plants belonging to its original species or family.

Molecular determinants of host jumps are largely unknown for plant viruses (but see [3] and references therein), in contrast with molecular determinants involved in the breakdown of resistance genes, i.e., host adaptation at the within-plant-species level. Perhaps the most striking result of this study is the large number of repeated correlated host gains in the potyvirus phylogenetic tree, i.e., the fact that several host gains were inferred on the same branch of the tree. It is questionable whether the molecular events involved in these different host changes are independent or not. Many of these correlated host gains occurred several times in the potyvirus phylogenetic tree and the frequency of such recurrent events largely outweighs the expectations of chance. Hence, it is probable that several of these host gains were not independent and that the acquired capacity to infect a first plant species increased the probability that the virus also acquired the capacity to infect a second plant species. Underlying molecular mechanisms could be either pleiotropy, i.e., the fact that a mutation is responsible for both host gains simultaneously, or the ‘springboard’ effect, where a mutation responsible for a first host gain favors the acquisition of secondary mutations involved in the second host gain. Recombination is also widespread in potyviruses and may alter their genome in such a drastic way that multiple traits could be affected simultaneously by a single recombination event. Likewise, in our dataset, one potyvirus species, WMV, is the result of interspecific recombination between *Soybean mosaic virus* (SMV) and BCMV [34] and has an expanded host range compared to its parental species (37 species compared to 19 and 21 species, respectively, for SMV and BCMV; Appendix A). One would expect that these molecular events would have affected the capacity to infect plant species belonging to the same family, since in general host range barriers for plant viruses are much more frequent and/or stronger for plants belonging to different families than within families [35]. An alternative “bridgehead” hypothesis would be that acquiring the capacity to infect a novel plant species allows a virus to invade a novel ecosystem and increases its exposure and capacity to jump to other plant species of that ecosystem [5]. In that case, we would expect that recurrent co-occurrences of host gains would preferentially affect plant species from the same geographical origin. Both the molecular and bridgehead hypotheses are supported by the fact that plant species corresponding to repeated host gain co-occurrences belong more frequently to the same botanical family or share more frequently the same continent of origin than expected under random.

## 5. Conclusions

This study shows that the potyvirus host range followed a dynamic pattern, with frequent gains or losses during evolution and, surprisingly, frequent and repeated co-occurrences of host gains in different potyvirus clades. It also confirms the potentially important role of agriculture and crop plants on the diversification of potyviruses. Consequently, potyvirus evolution and speciation in the future could be strongly affected by future agricultural changes, such as the choice of new crop species or of growing crop species in new geographical areas. Indeed, agricultural environments are favorable for virus adaptation, specialization, and speciation, given the large number and genetic homogeneity of hosts in crops [27,28,36].

The molecular determinants involved in potyvirus host jumps, either from the virus or the plant side, are largely unknown (but see [6,14,16]). Given the multiple virus and host factors interacting during virus infection [37,38,39], it would be interesting to test formally the hypothesis of common genetic determinants involved in multiple host gains in potyviruses by reverse genetics approaches. Our study suggests candidate potyviruses and plant species for these approaches (Table 3).

## Figures and Tables

**Figure 1 viruses-12-00111-f001:**
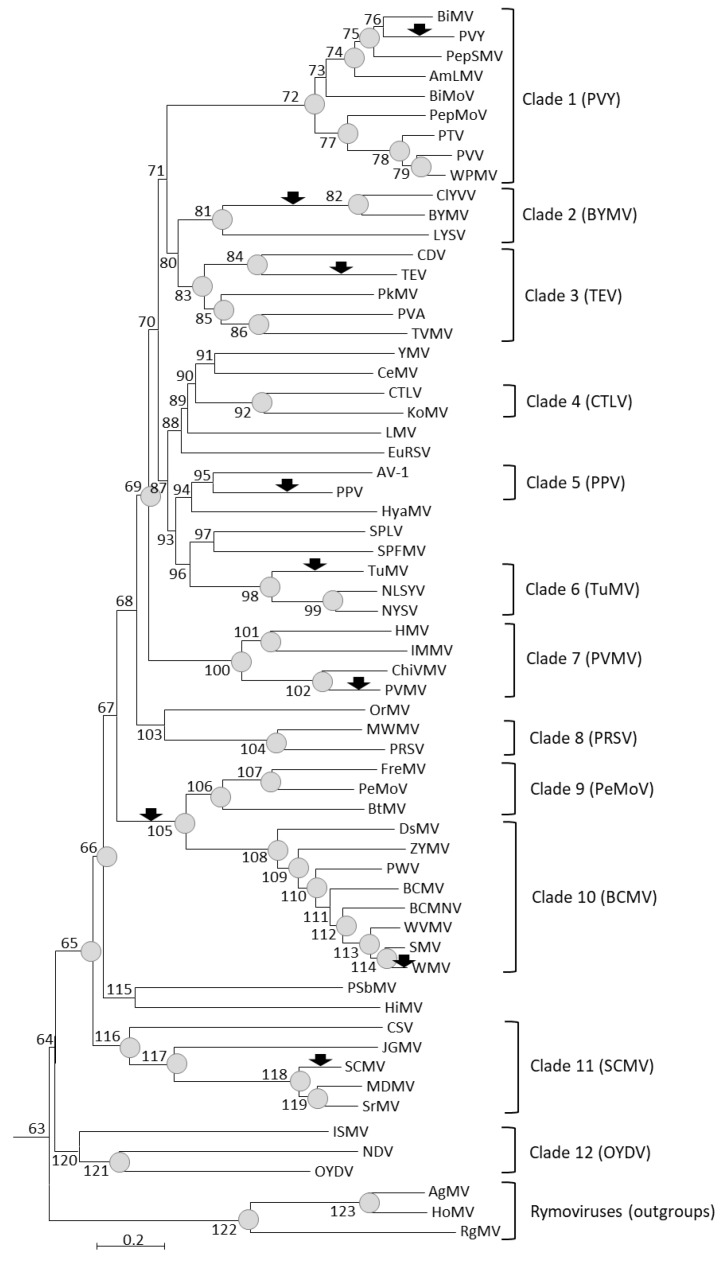
Phylogenetic tree of potyviruses and rymoviruses based on alignment of amino acid sequences of full-length genomes with MAFFT and the LG model with gamma-distributed rates among sites, invariant sites, and amino acid frequencies. Internal branches are numbered from 63 to 123. Terminal branches are named after the acronym of the corresponding virus species. Branches with bootstrap values above 0.7 are indicated by gray circles. Arrows indicate branches with a significantly higher number of host gains than expected by chance (Appendix A). Virus species corresponding to acronyms and GenBank accession numbers are as follows: *Agropyron mosaic virus* (AgMV; accession NC_005903), *Amaranthus leaf mottle virus* (AmLMV; MN709786), *Asparagus virus 1* (AV-1; NC_025821), *Bean common mosaic necrosis virus* (BCMNV; NC_004047), *Bean common mosaic virus* (BCMV; MK069988), *Bidens mottle virus* (BiMoV; NC_014325), *Bidens mosaic virus* (BiMV; NC_023014), *Beet mosaic virus* (BtMV; NC_005304), *Bean yellow mosaic virus* (BYMV; NC_003492), *Colombian datura virus* (CDV; NC_020072), *Celery mosaic virus* (CeMV; NC_015393), *Chilli veinal mottle virus* (ChiVMV; NC_005778), *Clover yellow vein virus* (ClYVV; NC_003536), *Cocksfoot streak virus* (CSV; NC_003742), *Carrot thin leaf virus* (CTLV; NC_025254), *Dasheen mosaic virus* (DsMV; MG602234), *Euphorbia ringspot virus* (EuRSV; NC_031339), *Freesia mosaic virus* (FreMV; NC_014064), *Hippeastrum mosaic virus* (HiMV; NC_017967), *Henbane mosaic virus* (HMV; MH779476), *Hordeum mosaic virus* (HoMV; NC_005904), *Hyacinth mosaic virus* (HyaMV; NC_037051), *Iris mild mosaic virus* (IMMV; JF320812), *Iris severe mosaic virus* (ISMV; NC_029076), *Johnsongrass mosaic virus* (JGMV; NC_003606), *Konjac mosaic virus* (KoMV; NC_007913), *Lettuce mosaic virus* (LMV; NC_003605), *Leek yellow stripe virus* (LYSV; NC_004011), *Maize dwarf mosaic virus* (MDMV; NC_003377), *Moroccan watermelon mosaic virus* (MWMV; NC_009995), *Narcissus degeneration virus* (NDV; NC_008824), *Narcissus late season yellows virus* (NLSYV; NC_023628), *Narcissus yellow stripe virus* (NYSV; NC_011541), *Ornithogalum mosaic virus* (OrMV; NC_019409), *Onion yellow dwarf virus* (OYDV; NC_005029), *Peanut mottle virus* (PeMoV; NC_002600), *Pepper mottle virus* (PepMoV; NC_001517), *Pepper severe mosaic virus* (PepSMV; NC_008393), *Pokeweed mosaic virus* (PkMV; NC_018872), *Plum pox virus* (PPV; NC_001445), *Papaya ringspot virus* (PRSV; NC_001785), *Pea seed-borne mosaic virus* (PSbMV; NC_001671), *Peru tomato mosaic virus* (PTV; NC_004573), *Potato virus A* (PVA; NC_004039), *Pepper veinal mottle virus* (PVMV; NC_011918), *Potato virus V* (PVV; NC_004010), *Potato virus Y* (PVY; NC_001616), *Passionfruit woodiness virus* (PWV; NC_014790), *Ryegrass mosaic virus* (RgMV; NC_001814), *Sugarcane mosaic virus* (SCMV; NC_003398), *Soybean mosaic virus* (SMV; NC_002634), *Sweet potato feathery mottle virus* (SPFMV; NC_001841), *Sweet potato latent virus* (SPLV; NC_020896), *Sorghum mosaic virus* (SrMV; NC_004035), *Tobacco etch virus* (TEV; NC_001555), *Turnip mosaic virus* (TuMV; KF595121), *Tobacco vein mottling virus* (TVMV; NC_001768), *Watermelon mosaic virus* (WMV; NC_006262), *Wild potato mosaic virus* (WPMV; NC_004426), *Wisteria vein mosaic virus* (WVMV; NC_007216), *Yam mosaic virus* (YMV; NC_004752), and *Zucchini yellow mosaic virus* (ZYMV; NC_003224).

**Table 1 viruses-12-00111-t001:** Summary results of phylogenetic inference of host range change.

Plant Species	Viruses with Host/Non-Host Status	LRT (*p* Value) ^1^	Potyvirus Ancestral State Inference ^2^	Permutation Test (*p* Value) ^3^	Non-Host to Host Changes ^4^	Host to Non-Host Changes ^4^	Number of Changes ^4^
*Allium cepa*	12	0.293	0.17	0.9	2	0	2
*Apium graveolens*	16	0.132	0.05	0.82	2	0	2
*Avena sativa*	14	0.824	0.36	0.56	2 *	0.5 *	2.5
*Beta vulgaris*	31	0.224	0.04	0.048	8	0	8
*Brassica campestris*	17	0.202	0.03	0.62	2	0	2
*Brassica oleracea*	23	0.303	0	0.56	2	0	2
*Capsicum annuum*	34	0.034	0.01	0.13	3.5 *	2.5 *	6
*Capsicum frutescens*	19	0.343	0.10	0.022	3	1	4
*Catharanthus roseus*	17	NA	0.16	0.93	4	0	4
*Chenopodium amaranticolor*	50	0.894	0.95	0.47	0	7.5 *	7.5
*Chenopodium quinoa*	51	0.38	0.91	0.70	0	9	9
*Citrullus lanatus*	15	0.77	0.20	0.017	3	0	3
*Cucumis melo*	19	0.48	0.04	0.15	3	0	3
*Cucumis sativus*	47	0.043	0.04	0.13	9	1	10
*Datura stramonium*	48	0.090	0.01	0.3	6	0	6
*Glycine max*	28	0.431	0.13	0.016	4	1	5
*Hordeum vulgare*	15	0.074	0.06	0.58	2	0	2
*Lactuca sativa*	22	0.047	0.08	0.83	4	0	4
*Lathyrus odoratus*	14	0.189	0.68	0.61	0	3	3
*Lupinus albus*	13	0.363	0.96	0.94	0	1	1
*Medicago sativa*	18	0.179	0.10	0.25	3	0	3
*Nicotiana benthamiana*	31	0.864	0.70	0.96	0	6.5 *	6.5
*Nicotiana clevelandii*	48	0.857	0.82	0.88	0	11	11
*Nicotiana glutinosa*	51	0.146	0.04	0.003	11	2	13
*Nicotiana megalosiphon*	22	0.705	0.33	0	2	1.5 *	3.5
*Nicotiana sylvestris*	13	0.164	0.93	0.81	0	2	2
*Nicotiana tabacum*	53	0.167	0.33	NA	NA	NA	NA
*Phaseolus vulgaris*	49	0.413	0.15	0.11	7.5 *	2	9.5
*Raphanus sativus*	20	0.2	0	0.42	1	0	1
*Solanum lycopersicum*	43	0.201	0.09	0.039	8	0	8
*Solanum melongena*	20	0.040	0.02	0.43	3	0	3
*Solanum tuberosum*	20	NA	0.19	0.29	5	0	5
*Trifolium pratense*	22	0.082	0	0.013	3	0	3
*Trifolium repens*	20	0.452	0.04	0.6	2	0	2
*Trigonella foenum-graecum*	14	0.186	0.92	0.93	0	2	2
*Triticum aestivum*	15	0.657	0.19	0.59	1	0	1
*Vicia faba*	33	0.385	0.09	0.004	5	2	7
*Vigna unguiculata*	44	0.131	0.06	0.26	6.5 *	1	7.5
*Zea mays*	25	0.742	0	0.046	1	0	1
Average per plant species	-	-	-	-	3.12	1.49	4.61

^1^ LRT: Likelihood ratio test of models all-rates-different (ARD) and equal-rates (ER); *p* < 0.05 means that model ARD is preferred over model ER. ^2^ Probability *P*_64_ that the plant species was host of the potyvirus most recent common ancestor (MRCA; branch number 64 in Figure 1). For *Avena sativa*, *P*_64_ was above the 0.35 threshold (0.36), but the status of next branch in the tree was inferred with more confidence (*P*_65_ = 0.24) allowing reliable host change reconstruction. ^3^ The *p* value indicates the frequency of random permutations of host and non-host data among the potyvirus species providing a probability that the plant species was host of the potyvirus MRCA lower than the inferred one (*P*_64_ in column 4). ^4^ Number of changes among potyviruses (i.e., excluding rymoviruses). * Reconstruction of host range evolution was uncertain in some parts of the virus phylogenetic tree. Consequently, two alternative scenarios of host range evolution were considered, and the indicated number of changes corresponds to the average of these two scenarios. NA: Not available (for LRT, it means that model ARD did not converge).

**Table 2 viruses-12-00111-t002:** Distribution of the analyzed plant species among botanical families and crop or wild plant categories.

Family	Crop	Wild	Total
Amaranthaceae	2	1	3
Poaceae	4	0	4
Brassicaceae	3	0	3
Solanaceae	6	6	12
Cucurbitaceae	3	0	3
Fabaceae	10	0	10
Others	3	1	4

**Table 3 viruses-12-00111-t003:** Host gain co-occurrence (i.e., host gains inferred on the same branch of the potyvirus tree) found repeatedly (i.e., on at least two different branches for a given pair of plant species).

Plant Species 1	Plant Species 2	Branch Name or Number (Figure 1)	Independent Host Gain Co-Occurrences	Plant Species 1 and 2 Belong to the Same Family	Plant Species 1 and 2 Belong to the Same Continent of Origin ^1^
*Beta vulgaris*	*Brassica campestris*	PSbMV, TuMV	2	NO	YES (EU)
*Beta vulgaris*	*Brassica oleracea*	PSbMV, TuMV	2	NO	YES (EU)
*Beta vulgaris*	*Cucumis sativus*	PVY, TuMV, 82	3	NO	YES (EU)
*Beta vulgaris*	*Lactuca sativa*	LMV, TuMV	2	NO	YES (EU)
*Beta vulgaris*	*Medicago sativa*	PSbMV, 82	2	NO	YES (EU)
*Beta vulgaris*	*Nicotiana glutinosa*	BtMV, TEV, TuMV, 82	4	NO	NO
*Beta vulgaris*	*Phaseolus vulgaris*	LMV, 82	2	NO	NO
*Beta vulgaris*	*Solanum lycopersicum*	PVY, TEV	2	NO	NO
*Beta vulgaris*	*Solanum melongena*	PVY, TEV	2	NO	YES (EU)
*Beta vulgaris*	*Solanum tuberosum*	PVY, TEV	2	NO	NO
*Beta vulgaris*	*Vicia faba*	PSbMV, TuMV, 82	3	NO	YES (EU)
*Beta vulgaris*	*Vigna unguiculata*	PVY, 82	2	NO	NO
*Brassica campestris*	*Brassica oleracea*	PSbMV, TuMV	2	YES	YES (EU)
*Brassica campestris*	*Vicia faba*	PSbMV, TuMV	2	NO	YES (EU)
*Brassica oleracea*	*Vicia faba*	PSbMV, TuMV	2	NO	YES (EU)
*Capsicum annuum*	*Solanum lycopersicum*	TEV, 100	2	YES	YES (AM)
*Capsicum annuum*	*Solanum tuberosum*	TEV, 100	2	YES	YES (AM)
*Capsicum frutescens*	*Nicotiana glutinosa*	TEV, 72	2	YES	YES (AM)
*Catharanthus roseus*	*Datura stramonium*	PVMV, WMV	2	NO	NO
*Catharanthus roseus*	*Nicotiana glutinosa*	PVMV, WMV	2	NO	NO
*Citrullus lanatus*	*Cucumis sativus*	82, 104	2	YES	NO
*Citrullus lanatus*	*Glycine max*	82, 105	2	NO	NO
*Citrullus lanatus*	*Phaseolus vulgaris*	82, 105	2	NO	NO
*Citrullus lanatus*	*Vigna unguiculata*	82, 105	2	NO	YES (AF)
*Cucumis melo*	*Medicago sativa*	BYMV, 108	2	NO	NO
*Cucumis sativus*	*Datura stramonium*	HiMV, TuMV, WMV	3	NO	NO
*Cucumis sativus*	*Nicotiana glutinosa*	HiMV, PPV, TuMV, WMV, 82	5	NO	NO
*Cucumis sativus*	*Solanum lycopersicum*	HiMV, PPV, PVY	3	NO	NO
*Cucumis sativus*	*Trifolium pratense*	PPV, WMV, 82	3	NO	YES (EU)
*Cucumis sativus*	*Trifolium repens*	PPV, 82	2	NO	YES (EU)
*Cucumis sativus*	*Vicia faba*	TuMV, 82	2	NO	YES (EU)
*Cucumis sativus*	*Vigna unguiculata*	PeMoV, PPV, PVY, 82, 109	5	NO	NO
*Datura stramonium*	*Nicotiana glutinosa*	HiMV, HMV, PVMV, TuMV, WMV	5	YES	YES (AM)
*Glycine max*	*Phaseolus vulgaris*	SCMV, 82, 105	3	YES	NO
*Glycine max*	*Vicia faba*	82, 105	2	YES	YES (EU)
*Medicago sativa*	*Vicia faba*	PSbMV, 82	2	YES	YES (EU)
*Nicotiana glutinosa*	*Solanum lycopersicum*	HiMV, PPV, TEV, 86	4	YES	YES (AM)
*Nicotiana glutinosa*	*Solanum melongena*	PVMV, TEV	2	YES	NO
*Nicotiana glutinosa*	*Trifolium pratense*	PPV, WMV, 82	3	NO	NO
*Nicotiana glutinosa*	*Trifolium repens*	PPV, 82	2	NO	NO
*Nicotiana glutinosa*	*Vigna unguiculata*	PPV, 82	2	NO	NO
*Phaseolus vulgaris*	*Vicia faba*	82, 105	2	YES	NO
*Solanum lycopersicum*	*Solanum melongena*	PVY, TEV	2	YES	NO
*Solanum lycopersicum*	*Solanum tuberosum*	PVY, TEV, 100	3	YES	YES (AM)
*Solanum lycopersicum*	*Vigna unguiculata*	PPV, PVY	2	NO	NO
*Solanum melongena*	*Solanum tuberosum*	PVY, TEV	2	YES	NO
*Trifolium pratense*	*Trifolium repens*	PPV, 82	2	YES	YES (EU)
*Trifolium pratense*	*Vigna unguiculata*	PPV, 82	2	YES	NO
*Trifolium repens*	*Vigna unguiculata*	PPV, 82	2	YES	NO
Total			119		

^1^ EU: Eurasia and Mediterranean Basin; AF: Africa; and AM: Americas.

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
