# Peer review of "Host Range Evolution of Potyviruses: A Global Phylogenetic Analysis"

_viruses, 2020, doi:10.3390/v12010111_

Round 1

Reviewer 1 Report

This manuscript presents interesting data on which to unravel the evolution of host species ranges in the potyviruses using analysis of phylogenetic relationships between 62 virus species for which genome and host range. The effort alone deserves credit, but this is combined with an excellent presentation of the data and a very well-written text. This paper is recommended for publication.

Author Response

Thanks a lot!

Reviewer 2 Report

1.Introduction:  "The genus Potyvirus is quite recent, its diversification having begun circa 7000 years ago, i.e. at the dawn of agriculture [19]." This is an interesting comment and an additional sentence to explain further would be helpful. 

2. Is there any way to make predictions of where future potyvirus-host evolution may occur? Could a new breeding technology such as genome editing of certain crop species artificially direct that evolution?

Author Response

1.Introduction:  "The genus Potyvirus is quite recent, its diversification having begun circa 7000 years ago, i.e. at the dawn of agriculture [19]." This is an interesting comment and an additional sentence to explain further would be helpful.

We modified the Introduction accordingly :

Phylogenetic analyses showed that the diversification of the genus Potyvirus is quite recent,since it has begun circa 7000 years ago probably in the greater region around the Fertile Crescent of the Middle East, where agriculture emerged about 9500 years ago [19]. This suggests that potyviruses have undergone numerous host range changes in a relatively short amount of time and that crop plants have contributed substantially to the diversification of potyviruses.

Is there any way to make predictions of where future potyvirus-host evolution may occur?

We modified the Conclusion accordingly (as far as we could!):

« It also confirms the potentially important role of agriculture and crop plants on the diversification of potyviruses. Consequently, potyvirus evolution and speciation in the future could be strongly affected by future agricultural changes, such as the choice of new crop species or of growing crop species in new geographical areas. Indeed, agricultural environments are favourable for virus adaptation, specialization and speciation, given the large number and genetic homogeneity of hosts in crops [27,28,39].»

Could a new breeding technology such as genome editing of certain crop species artificially direct that evolution?

Answer : Yes maybe. As there is some (limited) evidence that host jump molecular mechanisms are similar to mechanisms involved in resistance breakdown [ref 14,16], it is possible that editing of single genes corresponding to « classical » virus resistance genes may also affect potyvirus host jumps and speciation. However, this is based on little evidence and highly speculative. So, we decided not to discuss that aspect, which is also quite far away from the article’s topic.